# Transition from Screw-Type to Edge-Type Misfit Dislocations at InGaN/GaN Heterointerfaces

**Quantong Li** [1,2,*], **Albert Minj** [3] , **Yunzhi Ling** [1], **Changan Wang** [1,4], **Siliang He** [5], **Xiaoming Ge** [1], **Chenguang He** [1] , **Chan Guo** [1], **Jiantai Wang** [1], **Yuan Bao** [1], **Zhuming Liu** [1,*] and **Pierre Ruterana** [2]

1     Institute of Semiconductors, Guangdong Academy of Sciences, Guangzhou 510650, China;
      lingyunzhi@gdisit.com (Y.L.); wangchangan@gdisit.com (C.W.); gexiaoming@gdisit.com (X.G.);
      hechenguang@gdisit.com (C.H.); guochan@gdisit.com (C.G.); wangjiantai@gdisit.com (J.W.);
      baoyuan@gdisit.com (Y.B.)
2     CIMAP UMR 6252, CNRS ENSICAEN UCBN CEA, 6 Boulevard du Maréchal Juin, 14050 Caen, France;
      pierre.ruterana@ensicaen.fr
3     Interuniversity Microelectronics Centre (IMEC), Kapeldreef 75, 3000 Leuven, Belgium; albert.minj@imec.be
4     School of Electronics & Communication, Guangdong Mechanical and Electrical Polytechnic,
      Guangzhou 510515, China
5     Key Laboratory of Microelectronic Packaging & Assembly Technology of Guangxi Education Department,
      School of Mechanical & Electrical Engineering, Guilin University of Electronic Technology,
      Guilin 541004, China; siliang_he@guet.edu.cn
*     Correspondence: lqtnano@163.com (Q.L.); liuzhuming@gdisit.com (Z.L.)

**Abstract:** We have investigated the interface dislocations in $In_xGa_{1-x}N/GaN$ heterostructures ($0 \leq x \leq 0.20$) using diffraction contrast analysis in a transmission electron microscope. The results indicate that the structural properties of interface dislocations depend on the indium composition. For lower indium composition (up to $x = 0.09$), we observed that the screw-type dislocations and dislocation half-loops occurred at the interface, even though the former do not contribute toward elastic relaxation of the misfit strain in the InGaN layer. With the increase in indium composition ($0.13 \leq x \leq 0.17$), in addition to the network of screw-type dislocations, edge-type misfit dislocations were generated, with their density gradually increasing. For higher indium composition ($0.18 \leq x \leq 0.20$), all of the interface dislocations are transformed into a network of straight misfit dislocations along the <10–10> direction, leading to partial relaxation of the InGaN epilayer. The presence of dislocation half-loops may be explained by a slip on basal plane; formation of edge-type misfit dislocations are attributed to punch-out mechanism.

**Keywords:** InGaN/GaN heterostructures; transmission electron microscopy; indium composition; screw-type dislocations; edge-type misfit dislocations





## 1. Introduction

The direct bandgap of InGaN alloys has been attracting tremendous research attention, due to their excellent emission properties for light-emitting diodes (LEDs) and laser diodes (LDs) [1,2]. However, several critical issues related to epitaxial growth remain unsolved [3]. This is because of a larger lattice mismatch of up to ~11% between GaN and InN, which results in a compositional instability in InGaN [4]. Therefore, solving the lattice mismatch has become an important prerequisite for high-quality InGaN epilayer growth, and it is urgently needed to achieve high efficiency and high luminescence in green light-emitting devices [3]. In the process of growing InGaN films on GaN substrates, large lattice mismatches lead to huge strains, resulting in plastic relaxation of the misfit strain. This has been evidenced as an occurrence of parallel networks of straight misfit dislocations aligned along the <10–10> orientation in compressively strained InGaN/GaN [5–9], as well as in tensile-strained AlGaN/GaN [10] heterostructures. III-nitride films grown on GaN templates are short of the main slip planes, due to the absence of resolved shear

stress, and the plastic strain relaxation requires activation of the secondary slip plane. The beginning of misfit dislocation is attributed to intrinsic and external factors; for example, the formation of growing surface steps [11,12], adoption of epitaxial lateral overgrowth GaN templates [5], and pre-existing threading dislocations in GaN templates as reported by Mathews [13–16]. Different models based on force balance [17] or energy equilibrium equations [18], including that of Fischer and Blakeslee-Matthews [13,19], are used to account for the generation of misfit dislocations via the slip process. Formation of a misfit dislocation is first attributed to its nucleation at the surface, followed by its glide to the heterointerface on {11–22} pyramidal planes [5]. In this study, we found that, in the strain-relaxation process, only a part of heterostructures are fully strained, while the other parts are partially relaxed. In this paper, we have investigated the transition from screw-type to edge-type misfit dislocations in $In_xGa_{1-x}N$/GaN heterostructures ($0 \leq x \leq 0.20$) grown on (0001) sapphire templates.

## 2. Materials and Methods

A series of metal-organic chemical vapor depositions grown InGaN/GaN heterostructures with varying indium compositions were used in this study. The InGaN layers were grown at temperatures of 700–730 °C. The varying indium compositions of InGaN epilayers were confirmed by X-ray diffraction. The thickness of corresponding InGaN epilayers was measured between 40 and 110 nm by cross-sectional transmission electron microscope (TEM) imaging. Structural properties of interface dislocations were investigated by dark-field TEM diffraction contrast analysis, and by high-resolution TEM. Here, plan-view specimens used in the TEM analysis were prepared by the following steps: tripod wedge mechanical polishing of the samples to 12 μm, then ion milling (Gatan PIPS) at −40 °C with a 5 keV $Ar^+$ gun, incident to the sample surface, stopping ion milling when electronic transparency was reached. The observations along the [0001] zone axis were performed in a TEM operating at 200 kV.

## 3. Results and Discussions

Without indium composition, no defects were found in plan-view samples (not shown here). Figure 1a,b exhibit the [0001] zone axis and [1–100] dark field plan-view TEM images of $In_{0.09}Ga_{0.91}N$/GaN heterostructure, respectively. A network of straight dislocation lines and dislocation half-loops can be seen in Figure 1a. These dislocation lines extend over several micrometers in the entire observable area, with 100 nm of average inter-line spacing. These dislocation lines are aligned along the <11–20> direction, which can be assessed from the zone axis [0001] diffraction condition in the inset of Figure 1a. The Burgers vectors (***b***) of three sets of dislocations can be determined by a TEM diffraction contrast analysis [20]. As shown in the dark field image (see Figure 1b) taken under ***g*** = [1–100] diffraction conditions from the same area, we observe that only these dislocations along [–1–120] direction (yellow arrow) disappear, indicating that this set of dislocations satisfy the ***g***. ***b*** = 0 criterion of invisibility. Therefore, ***b*** of this set of dislocations is ***b*** = ***a*** = 1/3[–1–120], which is parallel to the dislocation line orientation, and are determined to be of the screw type. The other two sets of dislocations are also verified as screw ones according to the same method. Thus, all the interface dislocations in $In_{0.09}Ga_{0.91}N$/GaN heterostructures are determined as pure screw dislocations, with ***b*** = ***a***. The occurrence of screw dislocations is unusual as it well known that the screw-type dislocations do not relax the strain of the epilayer.

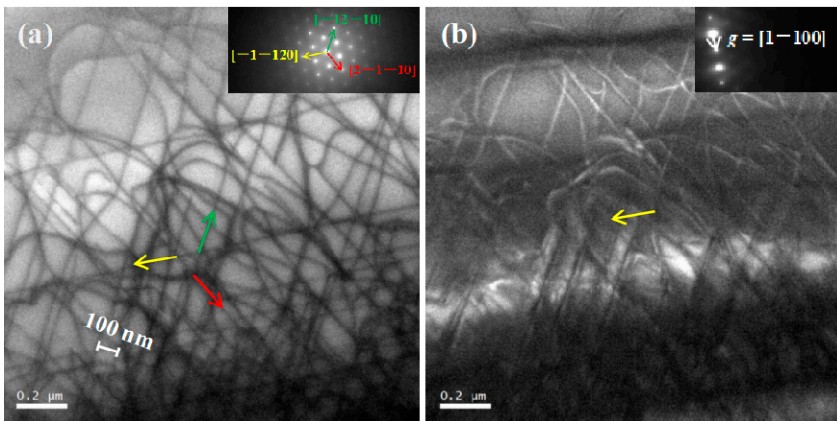

**Figure 1.** (**a**) Plan-view image and (**b**) plan-view weak beam dark-field image of $In_{0.09}Ga_{0.91}N/GaN$ heterostructure. Insets show the diffraction condition along the [0001] zone axis and the diffraction condition $g$ = [1–100], respectively.

Additionally, for a higher In-content of 17%, a network of dislocation lines with an average interline spacing of 200 nm can be seen in the dark-field images obtained at $g$ = [10–10] (see Figure 2a). These dislocations are aligned along the <11–20> direction, and two sets of dislocations are aligned along the <10–10> direction. The dark field image for $g$ = [10–10] of the same area (Figure 2b) shows that the dislocations along the [–12–10] direction (green arrow) are invisible. Hence, $b$ = $a$ = 1/3[–12–10] for these dislocations, and the direction is parallel to the dislocation line. Therefore, this set of dislocations is considered to be of the screw type. Similarly, the other two sets of dislocations (yellow and red arrows) along the equivalent <11–20> directions are also verified as screw-type ones. We also found that the dislocations along the [10–10] direction (blue arrow) also become invisible under this beam condition, which means that $b$ = $a$ = 1/3[–12–10], i.e., it is perpendicular to the direction of the dislocation line. Hence, this set of dislocations are identified as edge dislocations, i.e., misfit dislocations (MDs). The other set of dislocations (orange arrow) along the equivalent <10–10> direction is also verified as MDs. Thus, for In = 17%, all the observed dislocations have a Burgers vector of $a$, with most of them being screw dislocations along the <11–20> direction, and the remaining being MDs along the <10–10> direction.

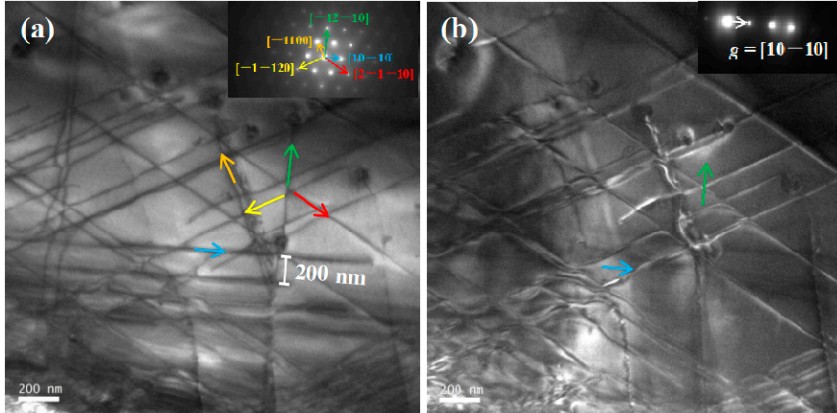

**Figure 2.** (**a**) Plan-view image and (**b**) plan-view weak beam dark-field image of $In_{0.17}Ga_{0.83}N/GaN$ heterostructure. Insets show the diffraction condition along the [0001] zone axis and the diffraction condition $g$ = [10–10], respectively.

When there is a higher In-content (In = 20%), only one type of dislocation network can be seen in bright filed TEM image (see Figure 3a), unlike the previous two cases. The dislocations are aligned along the <10–10> direction, with average interline spacing

of ~200 nm. In Figure 3b, we observe that the dislocations along the [0–110] direction (yellow arrow) become invisible in the dark field image obtained when **g** = [01–10]. Thus, **b** = **a** = 1/3[2–1–10] for these dislocations, which is perpendicular to its line direction, again classifying them as MDs. The other two sets of dislocations (green and red arrows) are also verified as MDs. Thus, all the observed dislocations are pure MDs, which contributes to the relaxation of the lattice mismatch strain. A part of the dislocations were observed to dissociate into two individual dislocations, which align along the <10–10> direction, as displayed in Figure 3. Diffraction results displayed that the individual dislocations were also of the a-type ones. This dissociation indicates that the MDs at the **interface** exist in Burgers vectors of both **a** and 2**a**, as reported by Liu and Li et al. [6,8].

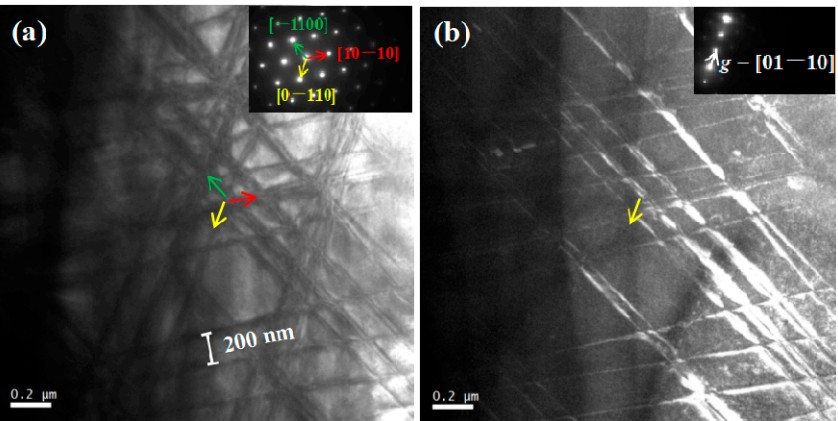

**Figure 3.** (**a**) Plan-view image and (**b**) plan-view weak beam dark-field image of $In_{0.2}Ga_{0.8}N/GaN$ heterostructure. Insets show the diffraction condition along the [0001] zone axis and the diffraction condition $g$ = [01–10], respectively.

One can observe a particular trend in the occurrence of dislocations from Figures 1–3. As the lattice mismatch increases, with an indium composition ranging from 9% to 20%, the interface dislocations transform from entirely screw dislocations into entirely misfit dislocations. In sample $In_{0.13}Ga_{0.87}N$, this transformation process is well captured. Figure 4 displays a bright-field plan-view TEM image, which clearly exhibits this transformation process of the dislocations from screw-type to edge-type; detailed analysis is shown in a schematic diagram below. This implies plastic relaxation of the layers gradually replaces elastic relaxation process.

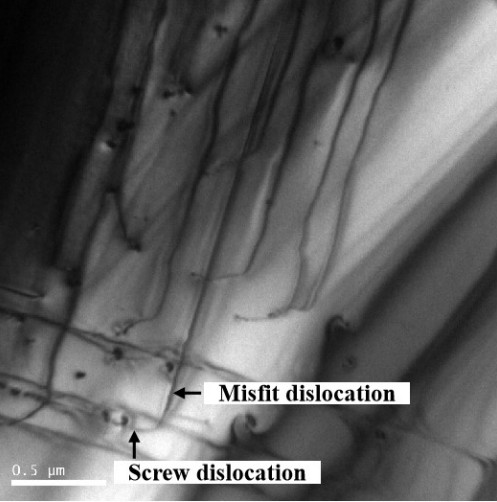

**Figure 4.** Weak beam bright field plan-view TEM image of $In_{0.13}Ga_{0.87}N$, taken under the diffraction condition g = [10–10].

To determine that the dislocations in the plan-view TEM images are interface dislocations, a high-resolution TEM (HRTEM) image of $In_{0.19}Ga_{0.81}N/GaN$ heterostructure was taken under the [1010] zone axis, as displayed in Figure 5a. It can be observed that the plan-view dislocations exist at the heterointerface (marked with the dashed line). Because of the large lattice misfit strain, the atomic arrangement in the heterostructure cannot be clearly seen in the HRTEM image. For ease of observation of the {1120} lattice fringes, we use in-plane Fourier points to obtain a Fourier filtered image. Figure 5b is a Fourier filtering image of Figure 5a. We clearly observed in Figure 5b that an extra half plane of atoms lie at the interface, therefore these interface dislocations are MDs (see the dashed line).

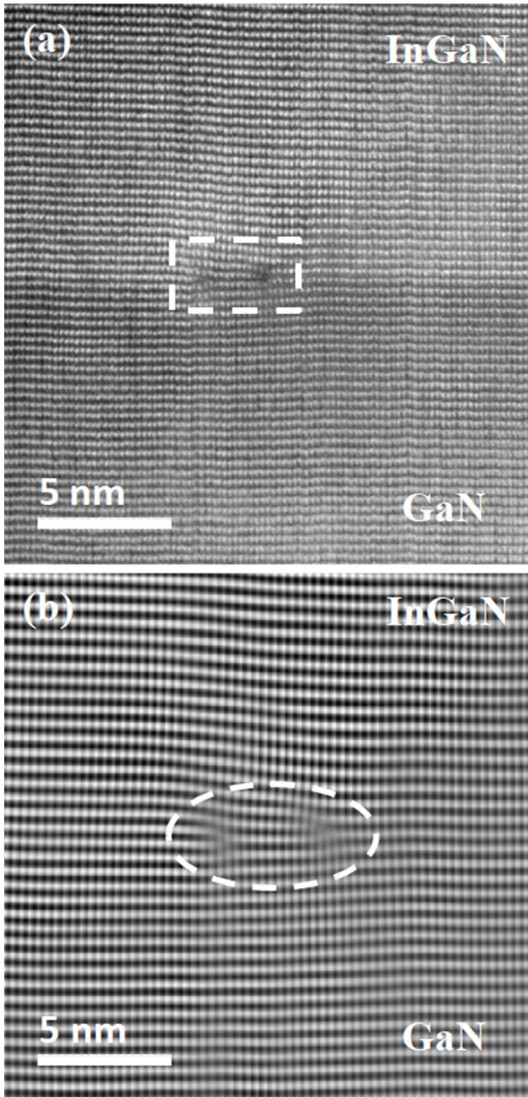

**Figure 5.** (**a**) High-resolution TEM image of $In_{0.19}Ga_{0.81}N/GaN$ heterostructure along the [1010] zone axis, (**b**) Fourier filtering image of the heterostructure.

It is commonly understood that the strain occurring because of lattice mismatch between epitaxial layers and substrates is eased to an extent either by elastic relaxation, or by plastic relaxation. Strain relaxation mechanisms found in the InGaN/GaN system have been reported by Liu et al. [21], see Figure 6. Figure 6a displays a diagram explaining the formation process of the dislocation half-loops. At the intersection of the free surface and the heterointerface, the atomic bonds are rearranged to form a ledge at the free surface, because of the mismatched shear stress. Nucleation and propagation of the dislocation half-loop happens at the corner of the pit, during the process driven by the mismatch

stress. As shown in Figure 6b, the half-loop is segmented into three line segments: $l_1$, $l_2$ and $l_3$. The Burgers vectors ($b_1 = b_3 = a = 1/3<11–20>$) of the dislocations $l_1$ and $l_3$ are parallel to the dislocation line direction of $<11–20>$, thus, $l_1$ and $l_3$ are identified as screw-type dislocations (elastic relaxation). The Burgers vector ($b_2 = a = 1/3<11–20>$) of the dislocations $l_2$ is perpendicular to the dislocation line direction of $<10–10>$, $l_2$ is thus an edge-type dislocation (plastic relaxation), i.e., a misfit dislocation. This mechanism is illustrated by the schematic diagrams shown in Figure 6d,e. Further relief of the misfit strain happens by continuous formation of misfit dislocations. The Burgers vector is in the glide plane; therefore, propagation of the half-loops occurs by a slip on the basal plane.

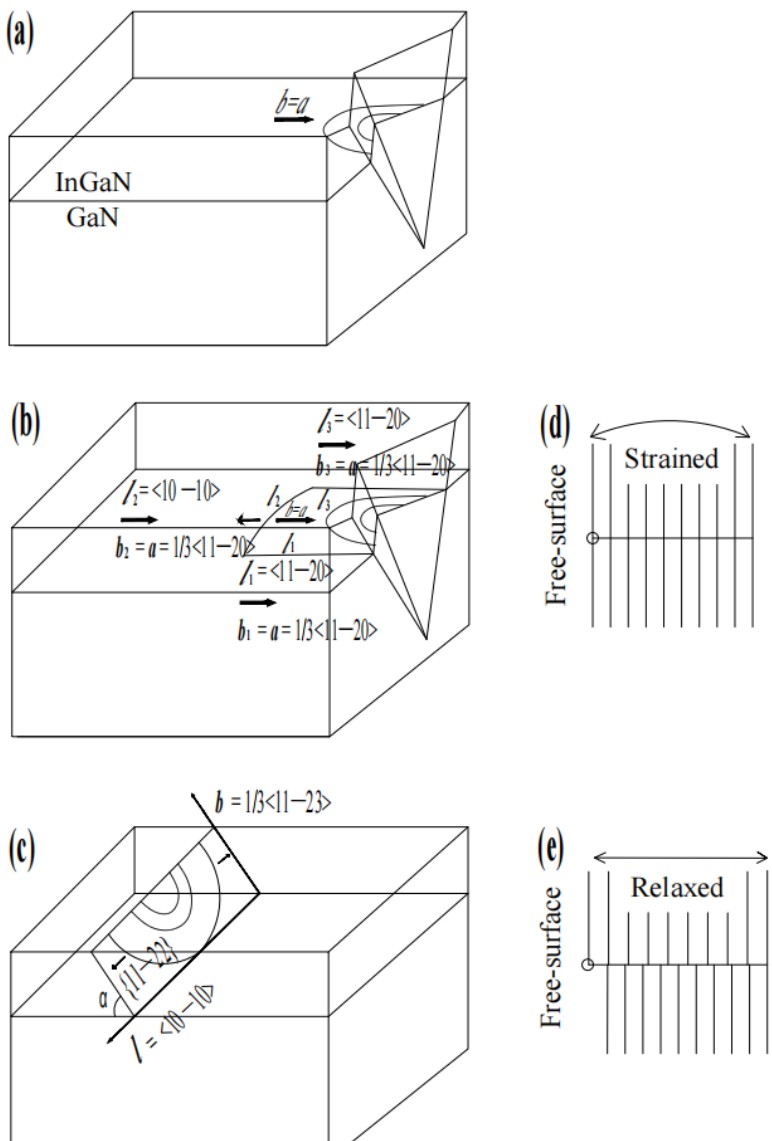

**Figure 6.** Schematic diagrams illustrating formation mechanisms of screw-type and edge-type misfit dislocations in the InGaN/GaN heterostructure. (**a**) Generation of a dislocation half-loop from a pit. (**b**) Transformation of a dislocation half-loop into screw dislocations ($l_1$ and $l_3$) and misfit dislocations ($l_2$) with $b = a$. (**c**) Glide of dislocations via the slip system $\{11–22\}<11–23>$, and formation of a misfit dislocation with $b = a + c$. (**d**) Strained state before critical indium composition. (**e**) Relaxed state after critical indium composition.

Due to a shortness of primary slip planes in the absence of resolved shear stresses in InGaN films grown on GaN (0001), plastic strain relaxation requires the activation of

secondary glide planes, primary pyramidal ones. As explained by Srinivasan et al. [5], only three of the pyramidal plane systems allows for non-zero resolved shear stress: {1–102}<1–101>, {11–22}<11–23> and {1–101}<11–23>. As they glide within one of the slip systems, three main forces are exerted on them: the lattice misfit stress $F_m$ is the driving force, and the line tension $F_l$ and the Peierls force $F_p$ are the resisting forces [22–24]. The net force for the half-loop slip is given by $F_{net} = F_m = F_l = F_p$. The pyramidal plane {1–102}<1–101> cannot contribute to the dislocation slip because of high Peierls force, while the pyramidal plane {11–22}<11–23> is more favorable than {1–101}<11–23> because of the large net driving force. It is reported that straight misfit dislocation lines can be formed at the InGaN/GaN heterointerface, via a slip or punch-out mechanism [6,8], as shown in Figure 6c. Nucleation and injection of the network of misfit dislocations <10–10> occur in pyramidal plane {11–22}<11–23> [5,6]. The driving force $F_d$ of this process is the decomposition shear stress on the inclined plane {11–22}, $F_d = F_m \cos\alpha$. Since the driving force for generating dislocation half-loops is larger, $F_d = F_m \cos 0 = F_m$. The resisting force on the basal plane is smaller than that on plane {11–22}. Thus, plastic relaxation, via dislocation half-loops introduced at the heterointerface, is easier than that from the growth surface. Whereas interface dislocation half-loops are constrained by the larger line tension, the two ends of the half-loops are fixed to the edge of the pit. The ends of the dislocation half-loops are free to move on the growth surface; therefore, the dislocation half-loops can easily propagate along the {11–22}<11–23> slip system, and form a misfit dislocation network at the interface.

## 4. Conclusions

We researched the formation of interface dislocations in $In_xGa_{1-x}N/GaN$ ($0 \leq x \leq 0.20$) heterostructures grown on (0001) sapphire substrates. Without indium composition, no defects are found in plan-view samples. For x = 0.09, a network of screw dislocations and dislocation half-loops are observed at the heterointerface. For $0.13 \leq x \leq 0.17$, both the network of the screw dislocations and misfit dislocations are found, with the density of the misfit dislocations increasing gradually. As the indium composition ($0.18 \leq x \leq 0.20$) increases further, all the screw dislocations transform into misfit dislocations, with their lines along the <10–10> direction. The generation of screw dislocations and misfit dislocations is due to glide of half-loops on the basal plane; the half-loops were generated from the heterointerface intersecting the free surface. For a higher indium composition ($x \geq 0.20$), the misfit dislocations were formed by the punch-out mechanism.

**Author Contributions:** Conceptualization, Q.L. and P.R.; methodology, Q.L. and P.R.; software, Q.L.; validation, Q.L. and Z.L.; formal analysis, C.G. and J.W.; investigation, Y.B.; resources, P.R. and Q.L.; data curation, Q.L. and C.H.; writing-original draft preparation, Q.L.; writing-review and editing, A.M., Y.L. and C.W.; visualization, S.H. and X.G.; supervision, Z.L. and P.R.; project administration, Q.L.; funding acquisition, Z.L. and Q.L. All authors have read and agreed to the published version of the manuscript.

**Funding:** This work was supported by Key-Area Research and Development Program of Guangdong Province (No. 2020B0101320002), the GDAS' Project of Science and Technology Development (No. 2021GDASYL-20210103074), and National Key Research and Development Program of China (No. 2022YFF0706100).

**Data Availability Statement:** Not applicable.

**Acknowledgments:** The authors thank the GDAS' Project of Science and Technology Development (No. 2020GDASYL-20200102024, 2021GDASYL-20210103077, 2019GDASYL-0103070 and 2022GDASZH-2022010111), the National Natural Science Foundation of China (No. 62101143 and 62104050), Key Area R&D Program of Guangzhou (No. 202103030002), Guangdong Basic and Applied Basic Research Foundation (No. 2021B1515120022 and 2022A1515110515), and the Science and Technology Program of Guangzhou (No. 201904020032). The authors would like to gratefully acknowledge the use of facilities at Centre de Recherche sur les Ions, les Matériaux et la Photonique at CNRS.

**Conflicts of Interest:** The authors declare no conflict of interest.

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
