# Peer review of "Transition from Screw-Type to Edge-Type Misfit Dislocations at InGaN/GaN Heterointerfaces"

_crystals, doi:10.3390/cryst13071027_

Round 1

Reviewer 1 Report

In this work, the generation of interfacial dislocations was investigated for InGaN/GaN heterostructures. The TEM diffraction contrast analysis showed that the dislocations were eventually changed from screw type to pure edge type when the indium content increased from 9% to 20%. Finally, a model for crystal relaxation was proposed to explain this behavior. I think this paper is suitable to be published in the journal of Crystals after correcting the following minor issues.

·        The terms “interfacial dislocation” and “interface dislocation” are used interchangeably throughout the text. It is better to use only one of them to keep the consistency.

·        Please specify which sample is used to obtain Fig. 5.

Author Response

Point 1: The terms “interfacial dislocation” and “interface dislocation” are used interchangeably throughout the text. It is better to use only one of them to keep the consistency.

Response 1: Only interfacial dislocation are used in the text. This is updated in the manuscript, please see the red font.

Point 2: Please specify which sample is used to obtain Fig. 5.

Response 2: The sample of In0.19Ga0.81N/GaN heterostructure is used to obtain Fig. 5. See the modified changes in the text:

“Figure 5. (a) High resolution TEM image of In0.19Ga0.81N/GaN heterostructure along [1010] zone axis, (b) Fourier filtering image of the heterostructure.”

Reviewer 2 Report

It was with great pleasure that I read the manuscript entitled 'Transition from screw type to edge type misfit dislocations at InGaN/GaN heterointerfaces' sent by the authors to your journal Crystals.
I believe that the value of the manuscript, in terms of acceptable presentation of results, is high.
I would like to ask the authors for some additional information:

1) In my opinion the authors should try to be more quantitative in the study of dislocations in heterostructures: I cite for example a work that could be of interest:

R. Gatti, F. Boioli, M. Grydlik, M. Brehm, H. Groiss, M. Glaser, F. Montalenti, T. Fromherz, F. Schäffler, Leo Miglio; Dislocation engineering in SiGe heteroepitaxial films on patterned Si (001) substrates. Appl. Phys. Lett. 21 March 2011; 98 (12): 121908. https://doi.org/10.1063/1.3569145

but there are many others in which theoretical physicists have tried to model dislocations in semiconductors.

2) Consequently, the discussion related to Figure 6 can be much more quantitative.

3) Can the authors perform spctroscopic analysis (e.g., light absorption, photoluminescence, steady state or time resolved) to study the photogenerated species with an ergetic scenario affected by the defects in the structure?

4) Can the authors do XPS/UPS analysis to study the energy levels?

5) Can the authors do Kelvin probe spectroscopy to study the energetic landscape?

Author Response

Point 1: In my opinion the authors should try to be more quantitative in the study of dislocations in heterostructures: I cite for example a work that could be of interest:

  1. Gatti, F. Boioli, M. Grydlik, M. Brehm, H. Groiss, M. Glaser, F. Montalenti, T. Fromherz, F. Schäffler, Leo Miglio; Dislocation engineering in SiGe heteroepitaxial films on patterned Si (001) substrates. Appl. Phys. Lett. 21 March 2011; 98 (12): 121908. https://doi.org/10.1063/1.3569145,

but there are many others in which theoretical physicists have tried to model dislocations in semiconductors.

Response 1: R. Gatti et al. reported a periodic trench dislocations in SiGe heteroepitaxial films, so it is suitable for quantitative study. Whereas the dislocations in InGaN/GaN heterostructure we studied were aperiodic, it is more suitable for qualitative study.

Point 2: Consequently, the discussion related to Figure 6 can be much more quantitative.

Response 2: Figure 6 schematically illustrates the generation process of aperiodic screw dislocations and misfit dislocations, this process is better for qualitative study.

Point 3: Can the authors perform spctroscopic analysis (e.g., light absorption, photoluminescence, steady state or time resolved) to study the photogenerated species with an ergetic scenario affected by the defects in the structure?

Response 3: The point is that the defects we are investigating are interfacial defects, the electronic and optical quality of the layers will mainly be determined by the crystalline quality of the bulk of the epitaxial layer, not the interface. The quality of the InGaN layer should improve with increase of layer thickness and the role of the interface defects if any should be decreased.

Point 4: Can the authors do XPS/UPS analysis to study the energy levels?

Response 4: We think that XPS or UPS can hardly be adequate for this system, they are surface techniques, and the interfaces we characterized are buried, so difficult to access with such techniques.

Point 5: Can the authors do Kelvin probe spectroscopy to study the energetic landscape?

Response 5: Line features in the three directions <10-10> can be seen the topography (the upper image). They originate at surface steps resulting from the slip planes. The corresponding contact potential difference (CPD) map obtained from KPFM analysis is shown in the bottom image. Here, increased CPD values by 0.10-0.49 V are seen over these line features, where the larger values are due to several misfit dislocations (MDs) located in the proximity and lowest value is for well separated individual MDs. Here the increase of CPD at MDs indicates presence of positive space charge region around MDs [Cavalcoli et al. Applied Surface Science, 2020, vol. 515, article 146016.]. In previous work, Minj et al. [Applied physics letters 109, 221106 (2016)] observed stimulated emission from single InGaN(50 nm)/GaN heterostructures only in the absence of misfit dislocations, which is true for In<17%. This is because of the non-radiative nature of the dislocations, which doesn’t allow the exciton density to reach the Mott density required for stimulated emission in the InGaN layer.

Reviewer 3 Report

- To compare the dislocation change in samples with different In content, the inspection of the structural changes should be performed for identical diffraction conditions. In the present version, the authors apply for the x=0.09 the diffraction condition g=[1-100], for 0.17 – g=[10-10], for 0.20 - g=[01-10].

- To guide the reader, the average inter-line spacing, as provided in the text (100-200 nm), should be marked in Figures 1a, 2a, 3a.

- For the reader to better follow the presentation of the results, the crystallographic directions in the text should correspond to the ones depicted in Figures (insets), which not always is the case. For example, in the text “…The three sets of dislocation lines are aligned along with the <11-20> directions, which can be assessed from the selected area diffraction pattern for zone axis [0001] in the inset of Figure 1a.” on Page 2, however, in Figure 1a, we obtain [-1-120]. The same is for Figure 2a, where in the text “…three sets of dislocation lines are aligned along with the <11-20> directions…”, but the [11-20] direction is missing in the Figure. The authors should check it throughout the paper.

- “…two sets of dislocation lines are along with the <10-10> directions…” on Page 3, therefore both directions should be depicted in the inset of Figure 2a.

- Statements for x=0.09 “…The other two sets of dislocations are also verified as screw ones according to the same method.” on Page 2, for x=0.17 “…Similarly, the other two sets of dislocations (yellow and red arrows) along the equivalent <11-20> directions are also verified as screw-type ones…” on Page 3 and for x=0.20 “…The other two sets of dislocations (green and red arrows) are also verified as MDs…” on Page 4, should be rewritten containing the particular dislocation line directions and the corresponding diffraction conditions, where the invisibility was observed.

- To understand the origin and the development of the dislocations, the InGaN/GaN interface line should be drawn in Figure 4.

- To correspond the TEM images of the three In content cases with the bird-view picture in Figure 6, the main plane (diffraction) and dislocation line directions, and the Burger vector directions should be presented. Best to draw a bird-view picture for each of the cases.

No particular comments.

Author Response

Point 1: To compare the dislocation change in samples with different In content, the inspection of the structural changes should be performed for identical diffraction conditions. In the present version, the authors apply for the x=0.09 the diffraction condition g=[1-100], for 0.17 – g=[10-10], for 0.20 - g=[01-10].

Response 1: The x=0.09 the diffraction condition g=[1-100], for 0.17 – g=[10-10], for 0.20 - g=[01-10], these three diffraction conditions are identical crystal orientation family g=<10-10>, and in order to illustrate the problem more clearly, different diffraction conditions are used for each image.

Point 2: To guide the reader, the average inter-line spacing, as provided in the text (100-200 nm), should be marked in Figures 1a, 2a, 3a.

Response 2: We have marked, please see Figures 1a, 2a, 3a.

Point 3: For the reader to better follow the presentation of the results, the crystallographic directions in the text should correspond to the ones depicted in Figures (insets), which not always is the case. For example, in the text “…The three sets of dislocation lines are aligned along with the <11-20> directions, which can be assessed from the selected area diffraction pattern for zone axis [0001] in the inset of Figure 1a.” on Page 2, however, in Figure 1a, we obtain [-1-120]. The same is for Figure 2a, where in the text “…three sets of dislocation lines are aligned along with the <11-20> directions…”, but the [11-20] direction is missing in the Figure. The authors should check it throughout the paper.

Response 3: The crystallographic directions in the text is corresponding to the ones depicted in Figures, because <11-20> directions are crystal orientation family, which includes [11-20],  [-1-120], [-12-10], [1-210], [2-1-10], [-2110].

Point 4: “…two sets of dislocation lines are along with the <10-10> directions…” on Page 3, therefore both directions should be depicted in the inset of Figure 2a.

Response 4: Both directions have been depicted in the inset of Figure 2a, please see [-1100] (orange arrow) and [10-10] (blue arrow).

Point 5: Statements for x=0.09 “…The other two sets of dislocations are also verified as screw ones according to the same method.” on Page 2, for x=0.17 “…Similarly, the other two sets of dislocations (yellow and red arrows) along the equivalent <11-20> directions are also verified as screw-type ones…” on Page 3 and for x=0.20 “…The other two sets of dislocations (green and red arrows) are also verified as MDs…” on Page 4, should be rewritten containing the particular dislocation line directions and the corresponding diffraction conditions, where the invisibility was observed.

Response 5: Statements for x=0.09 “…The other two sets of dislocations are also verified as screw ones according to the same method.” on Page 2, for x=0.17 “…Similarly, the other two sets of dislocations (yellow and red arrows) along the equivalent <11-20> directions are also verified as screw-type ones…” on Page 3 and for x=0.20 “…The other two sets of dislocations (green and red arrows) are also verified as MDs…” on Page 4, have been rewritten as “containing the particular dislocation line directions and the corresponding diffraction conditions, where the invisibility was observed.” Please see bold font in the text.

Point 6: To understand the origin and the development of the dislocations, the InGaN/GaN interface line should be drawn in Figure 4.

Response 6: Figure 4 shows a plan-view TEM image, which only exhibits InGaN or GaN layer, InGaN/GaN interface line can be drawn in cross-sectional TEM image.

Point 7: To correspond the TEM images of the three In content cases with the bird-view picture in Figure 6, the main plane (diffraction) and dislocation line directions, and the Burger vector directions should be presented. Best to draw a bird-view picture for each of the cases.   

Response 7: The bird-view picture for x=0.09 and x=0.17 is drawn in the upper image, the bird-view picture for x=0.20 is drawn in the bottom image.

Round 2

Reviewer 2 Report

The authors improved the manuscript considerably. If there had been a few more refinements from an experimental point of view, perhaps the manuscript could have been of broader scope for an audience that includes all of semiconductor physics. But it is nevertheless a good work that can, in my humble opinion, be considered for publication in the journal.

Author Response

Point 3: Can the authors perform spctroscopic analysis (e.g., light absorption, photoluminescence, steady state or time resolved) to study the photogenerated species with an ergetic scenario affected by the defects in the structure?

Response 3: The point is that the defects we are investigating are interfacial defects, the electronic and optical quality of the layers will mainly be determined by the crystalline quality of the bulk of the epitaxial layer, not the interface. The quality of the InGaN layer should improve with increase of layer thickness and the role of the interface defects if any should be decreased.

Point 4: Can the authors do XPS/UPS analysis to study the energy levels?

Response 4: We think that XPS or UPS can hardly be adequate for this system, they are surface techniques, and the interfaces we characterized are buried, so difficult to access with such techniques.

Reviewer 3 Report

- Point 2:

To guide the reader, the average inter-line spacing, as provided in the text (100-200 nm), should be marked in Figures 1a, 2a, 3a.

Response 2:

We have marked, please see Figures 1a, 2a, 3a.

COMMENT: Not done. While copying the text to the image, the authors do not make guidance for the readers about where the 100-200 nm spacing is.

- Point 4:

“…two sets of dislocation lines are along with the <10-10> directions…” on Page 3, therefore both directions should be depicted in the inset of Figure 2a.

Response 4:

Both directions have been depicted in the inset of Figure 2a, please see [-1100] (orange arrow) and [10-10] (blue arrow).

COMMENT: Not done. The text should clearly state that guidance for the directions.

Point 7:

To correspond the TEM images of the three In content cases with the bird-view picture in Figure 6, the main plane (diffraction) and dislocation line directions, and the Burger vector directions should be presented. Best to draw a bird-view picture for each of the cases.

Response 7:

The bird-view picture for x=0.09 and x=0.17 is drawn in the upper image, the bird-view picture for x=0.20 is drawn in the bottom image.

COMMENT: Not done. The authors do not take into account the suggestions of the comment. Figure 6 is left poorly informative: main crystal plane directions (family), as in the dislocation analysis, aren’t attached/related to the bird-view structure, arrows are unlabeled, not all line segment directions are provided, etc. No bird-view picture in the bottom image.

Author Response

Point 2: 

To guide the reader, the average inter-line spacing, as provided in the text (100-200 nm), should be marked in Figures 1a, 2a, 3a.

Response 2:

We have marked, please see Figures 1a, 2a, 3a. 

COMMENT: Not done. While copying the text to the image, the authors do not make guidance for the readers about where the 100-200 nm spacing is.

Response comment: We have marked in Figures 1a, 2a, 3a. Corresponding to changes please see in the text.

Figure 1. (a) Plan-view image and (b) plan-view weak beam dark-field image of In0.09Ga0.91N/GaN heterostructure. Insets show the diffraction condition along [0001] zone axis and the diffraction condition g = [1-100], respectively.

Figure 2. (a) Plan-view image and (b) plan-view weak beam dark-field image of In0.17Ga0.83N/GaN heterostructure. Insets show the diffraction condition along [0001] zone axis and the diffraction condition g = [10-10], respectively.

Figure 3. (a) Plan-view image and (b) plan-view weak beam dark-field image of In0.2Ga0.8N/GaN heterostructure. Insets show the diffraction condition along [0001] zone axis and the diffraction condition g = [01-10], respectively.

Point 4: 

“…two sets of dislocation lines are along with the <10-10> directions…” on Page 3, therefore both directions should be depicted in the inset of Figure 2a.

Response 4: 

Both directions have been depicted in the inset of Figure 2a, please see [-1100] (orange arrow) and [10-10] (blue arrow).

COMMENT: Not done. The text should clearly state that guidance for the directions.

Response comment: The crystallographic directions <10-10> in the text is corresponding to the ones depicted in the inset of Figure 2a, because <10-10> directions are crystal orientation family, which includes the directions [-1100] and [10-10].

Figure 2. (a) Plan-view image and (b) plan-view weak beam dark-field image of In0.17Ga0.83N/GaN heterostructure. Insets show the diffraction condition along [0001] zone axis and the diffraction condition g = [10-10], respectively.

Point 7:

To correspond the TEM images of the three In content cases with the bird-view picture in Figure 6, the main plane (diffraction) and dislocation line directions, and the Burger vector directions should be presented. Best to draw a bird-view picture for each of the cases.

Response 7:

The bird-view picture for x=0.09 and x=0.17 is drawn in the upper image, the bird-view picture for x=0.20 is drawn in the bottom image.

COMMENT: Not done. The authors do not take into account the suggestions of the comment. Figure 6 is left poorly informative: main crystal plane directions (family), as in the dislocation analysis, aren’t attached/related to the bird-view structure, arrows are unlabeled, not all line segment directions are provided, etc. No bird-view picture in the bottom image.

Response comment: Main crystal plane directions (family), as in the dislocation analysis, have been attached/related to the bird-view structure, arrows are labeled, all line segment directions are provided. Corresponding to changes please see in the text.

Figure 6. Schematic diagrams illustrating formation mechanisms of screw type and edge type misfit dislocations in the InGaN/GaN heterostructure. (a) Generation of dislocation half-loop from a pit. (b) Transformation of dislocation half-loop into screw dislocations (l1 and l3) and misfit dislocations (l2) with b=a. (c) Glide of dislocations via slip system {11-22}<11-23> and formation of misfit dislocation with b=a+c. (d) Strained state before critical indium composition. (e) Relaxed state after critical indium composition.